# The Impact of the COVID-19 Pandemic on the Prevalence of Head Lice Infestation among Children Attending Schools and Kindergartens in Poland

**DOI:** 10.3390/jcm12144819

**Published:** 2023-07-21

**Authors:** Marcin Padzik, Gabriela Olędzka, Anita Gromala-Milaniuk, Ewa Kopeć, Edyta Beata Hendiger

**Affiliations:** 1Parasitology Laboratory, Department of Medical Biology, Medical University of Warsaw, Litewska 14/16, 00-575 Warsaw, Poland; gabriela.oledzka@wum.edu.pl (G.O.); gromala-anita@wp.pl (A.G.-M.); edyta.hendiger@wum.edu.pl (E.B.H.); 2MAMY Z GŁOWY Sp. z o.o., Raclawicka 29a, 02-601 Warsaw, Poland; ewa.kopec@mamyzglowy.pl

**Keywords:** head lice infestation, pediculosis capitis, COVID-19, schools, kindergartens

## Abstract

Pediculosis capitis predominantly affects child populations. During the COVID-19 pandemic, various types of social limitations such as social isolation and limitations on educational facilities’ functionalities were introduced worldwide, including in Poland. The impact of these pandemic restrictions on the prevalence of pediculosis capitis has not been extensively investigated so far. Existing data on this topic are limited mainly to indirect evaluation methods, such as online surveys or Internet analysis. In this study, we directly examined a cohort of over five thousand children to assess the influence of COVID-19 pandemic restrictions on the prevalence of head lice infestation among school and kindergarten attendees in Poland. Our findings demonstrate that the restrictions imposed during the COVID-19 pandemic led to a decrease in the number of children infested with head lice compared to pre-pandemic data. These results confirm that head lice infestations remain an ongoing epidemiological concern and warrant continued monitoring.

## 1. Introduction

Human head lice (*P. humanus capitis*) are ectoparasites of humans [1,2]. They are wingless, blood-sucking insects living a parasitic way of life on the human scalp during their whole life cycle. The life cycle of *P. humanus capitis* consists of the following: egg, three nymphal stages, and adult form. Nymphs, like the adult forms, live on human scalps and feed on blood at each stage of development [2,3,4,5].

The route of transmission of head lice is direct, from host to host, during close contact [6,7]. Due to their social lifestyle, the most vulnerable group to pediculosis capitis are children aged between 5 and 13 years old [1,8,9,10,11]. The most common symptom of head lice infestation is itching of the scalp and skin redness. However, depending on the intensity of infestation the following symptoms may also occur: pain, erythema, scaling and burning, secondary infections (bacterial and fungal), pyoderma, lymphadenopathy, conjunctivitis, and even fever. The infestation may also remain asymptomatic [12,13,14,15,16]. The most effective method for diagnosing head lice infestation is through the use of appropriate detection combing, which can be performed using either a dry combing technique or wet combing with conditioner. The eggs, also known as nits, are typically laid on the nape of the neck and behind the ears, although they can also be found throughout the scalp. The most used treatment for pediculosis capitis is permethrin 1% lotion or shampoo. There are also other non-insecticidal agents, including dimethicone and isopropyl myristate which show promise in the treatment of head lice infestations [17,18].

Pediculosis capitis among children is still an issue in many parts of the world. In the United States of America (USA), head lice infestations are the most common among kindergarten children, elementary school students, and members of households with children [13]. Although there is no reliable data on the number of people infested with head lice in the USA, it is estimated that the number of infestations among children aged 3 to 11 years old is between 6 and 12 million yearly [13,19]. In China, where 303 school-aged children were surveyed in 2004, 43 (14.2%) were affected by head lice [20]. In India, the prevalence of head lice was checked between 2002 and 2004. Of 150 children working and living in slums, 72 (48%) were infested. Of 940 children attending public elementary schools, 156 (16.59%) students had head lice [21]. In Argentina, in 2003, 1.370 children attending private and public elementary schools were checked and lice were detected in 842 cases (61.4%) [22]. In the Czech Republic, between 2004–2005, 531 children were tested and 127 (23.9%) of them had lice or nits on their heads [23]. 

In Poland, head lice infestation does not fall under sanitary surveillance, which implies that there is no mandatory requirement to report cases of pediculosis capitis to the National Sanitary Inspectorate. Consequently, the absence of a national data collection system hinders the estimation of the magnitude of the issue. The lack of systematic reporting and data collection for head lice infestations in Poland presents a challenge in understanding the prevalence and distribution of the problem, making it difficult to implement targeted interventions and monitor trends effectively.Based on direct examinations performed in 2009–2012 in 30 schools in Lublin Province, Poland the prevalence of pediculosis capitis in schoolchildren was estimated at 2.01%Hhigher prevalence was noted in rural schools than in urban schools. Children between 8 and 12 years old were most frequently infested [24]. 

Based on the indirect surveys performed so far, pediculosis capitis was reported in the majority of tested schools in Poland (87.5%), while the greatest number of cases was reported in children aged 6–9 (68%) [25]. Data from the study conducted between 1996 and 2000 in Lubelskie Voivodship, Poland, indicated that of 42,759 girls, only 682 were infested with head lice (1.59%), and of 52,394 boys, 252 (0.48%) were infested [26,27]. During the COVID-19 outbreak, various measures such as social distancing, isolation, quarantine, enhanced sanitation practices, and the closure of public places, including schools and kindergartens, were implemented worldwide to mitigate the spread of the disease. Similarly, in Poland, after the first case of COVID-19 was reported on 4 March 2020, the government swiftly adopted these measures to curb the progression of the pandemic. Regardless of the severity of the pandemic, all educational institutions were operating under similar sanitary regimes. Based on the biology of *Pediculus humanus capitis* and the implementation of measures like social distancing and the closure of educational institutions during the COVID-19 pandemic, we formulated a hypothesis that a reduction in direct contact among children would impact the prevalence of head lice infestations.

However, this topic has not been studied extensively either in Poland or in other countries yet. If any studies were performed on this topic, indirect methods of investigation such as surveys or Internet searching were used. For example, Internet analysis performed in the United Kingdom showed that there was a significant drop in Internet searching of the term “head louse” from March 2020 onwards, coinciding with COVID-19 restrictions, and a large difference between pre- and post-pandemic search volumes [28]. Finding these limitations in the data, we decided to analyze our infestation data collected during direct examinations to evaluate the influence of COVID-19 pandemic restrictions on the prevalence of head lice infestation among children attending schools and kindergartens in Poland. By directly examining a large cohort of children, we aimed to provide more reliable and accurate information on the relationship between pandemic restrictions and head lice infestation rates.

## 2. Materials and Methods

### 2.1. Organization and Conduct of the Study

Children’s examinations in primary schools and kindergartens were conducted by certified nurses between 24 September 2018 and 29 June 2021. The examinations lasted 1 day and were performed “for cause”, which means that they were planned by the management of the institution. Participation of all children in the examination was voluntary and supported by the consent given by their parents to the management of the educational facility. The study was carried out in 64 kindergartens and 28 elementary schools in Warsaw, Krakow, and Poznan. According to the decision n°. AKBE/99/2021 of the Bioethics Committee of the Medical University of Warsaw, this study did not require any additional favorable opinion as it does not constitute a medical experiment.

### 2.2. Characteristics of the Study Group

The study group consisted of children from 3 to 14 years old attending primary schools and kindergartens. The pandemic restrictions introduced were the same for all educational facilities in Poland, so there should be no differences caused by this factor presumably. For logistical and financial reasons, the study was conducted in three cities, with most of the data collected in Warsaw. To mitigate this uneven sample stratification, we compared the total pediculosis capitis prevalence at pre-pandemic schools/kindergartens to the total prevalence at schools/kindergartens during the pandemic and assessed it statistically.

### 2.3. Human Head Lice Infestation Verification Methods

The direct examinations were conducted by certified nurses equipped with specialized magnifying glasses and combs according to CDC guidelines. Each child’s scalp was visually examined and combed with an individual comb and then carefully examined under a magnifying glass. The presence of live lice and/or eggshells/nits alone was an accepted criterion for the identification of an infestation. A group of the same nurses performed all of the examinations included in this study.

### 2.4. Statistical Analysis

The statistical analysis was performed using the GraphPad https://www.graphpad.com/quickcalcs/contingency1/ (accessed on 29 May 2023) ) statistical package. The infestation prevalence depending on the period of the pandemic and the type of educational facility was compared with the 2-tailed Fisher’s exact test. The test probability at the level of *p* < 0.05 was considered significant.

## 3. Results

There were 3255 children from kindergartens and 1753 children from primary schools directly examined, which means that a total of 5008 children were evaluated.

The pre-COVID study in kindergartens covered the period from 24 September 2018 to 28 February 2020. During this time, 2060 children were screened, of whom 248 were infested with head lice. The COVID-19 pandemic period of the study began on 14 September 2020 and ended on 29 July 2021. During this period, 1195 children were screened, of whom 116 were infested with head lice.

The pre-COVID study in schools covered the period from 4 September 2019 to 10 March 2020. During this time, 1027 children were screened, of whom 107 were infested with head lice. The COVID-19 pandemic period, during which the research was conducted, began on 24 September 2020 and ended on 17 June 2021. During this period, 726 children were screened, of whom 39 children were infested with head lice.

The granular data are shown in Figure 1 and Figure 2 and Table 1 below.

The obtained results provide insights into how social restrictions introduced in Poland during the COVID-19 pandemic affected the prevalence of human head lice infestation among the tested child population. The presented results indicate that the infestation prevalence during the COVID-19 pandemic significantly changed compared to the pre-COVID-19 pandemic period. In kindergartens, the prevalence during the pre-COVID-19 period reached 12.03%. During the COVID-19 period the prevalence was reduced to 9.7%. Even clearer differences in the results were obtained in the population of schoolchildren, among which the prevalence changed from 10.41% to 5.37% only. Both differences were considered statistically significant.

## 4. Discussion

Head lice are primarily transmitted through direct contact between individuals, making it reasonable to hypothesize that measures taken to reduce the transmission of COVID-19 could also have an impact on the prevalence of head lice infestations.

A cross-sectional descriptive study where an online survey was used determined the effect of isolation on pediculosis capitis infestations during quarantine and before the pandemic in Argentina, Buenos Aires. Data from 1118 children obtained from 627 surveys were analyzed. The overall prevalence of head lice decreased significantly from 69.6% before the COVID-19 pandemic to 43.9% during the COVID-19 pandemic. In addition, head lice infestations were more effectively controlled in households with up to two children than in households with three or more children. The restrictions that were applied consisted of the interruption of regular activities, which directly affected contact between children and thus the spread of head lice infestation [29].

The situation in the USA was different from that in Argentina. A significant increase in the number of diagnosed cases of head lice infestation was observed during the lockdown period, in 2020. Head lice clinics in the USA reported an increase in the number of head lice infestations in the spring of 2020 by an average of 25% over pre-pandemic COVID-19 values. Head lice infestations affected entire families and were more severe during the pandemic due to prolonged cohabitation in households. The numbers reported are based on reports from 112 out of 200 clinics opened in April and May 2020. The stay-at-home order resulted in the spread of head lice infestation among households [30]. Other data collected during monthly indirect monitoring for lice infestations in adult and pediatric patients in North Carolina showed significant decreases in the cases and prescriptions for lice treatment in the pediatric and adult populations accordingly [31].

The effect of lockdown on the spread of head lice infestation has also been tested in France. A population-based study was conducted by the indirect method of monitoring the sale of anti-lice drugs. The study period was 5 years including periods before and during the COVID-19 pandemic. Analysis of the database of the healthcare science company IQVIA Pharmaone LMPSO, which covers 14,000 pharmacies in France, was utilized. A clear decrease in sales, up to 50%, of topical anti-lice drugs and a less significant decrease in sales of ivermectin was observed during the periods of national isolation. However, other treatment options for lice infestation such as combing were not explored in this study. Additionally, the anti-lice treatments sold online were not included in the database [32,33].

In Israel, data comparing pediculicide sale numbers before and during the COVID-19 pandemic were used, as a possible marker for changing the epidemiology of head lice during the COVID-19 pandemic. The results of this study suggest that the extended isolation of children due to the COVID-19 pandemic significantly influenced the infestation rate of children with head lice [34].

In the present study, we directly examined 5008 children from educational institutions. We revealed that the prevalence of infested children in kindergartens and schools decreased significantly during the COVID-19 outbreak period. The data presented in our study correlate with the results obtained by other authors and indicate that the restrictions applied during the COVID-19 pandemic had an influence on the head lice infestation prevalence among the studied children population. However, when analyzing the results and drawing conclusions, one should consider factors such as the study methodology used (direct or indirect examinations of the study population), the specificity of the tested population (adults or children and their socio-economic status), and the extent of restrictions introduced locally during the COVID-19 pandemic (full or partial lockdown and the extent of its impact on the direct contacts of peers within the tested population). The data on COVID-19 and pediculosis capitis presented by other authors and discussed in this study are based mainly on indirect methods of evaluation such as online surveys or the monitoring of prescriptions for lice treatment or sales of lice treatment medicaments. These methods of pediculosis capitis prevalence measurement are less reliable and loaded with more variables and bias elements than the direct examination that was utilized in our study. However, there are also limitations of our approach that we would like to bring forward. We did not choose the educational facilities randomly. Visits to the schools and kindergartens were performed “for cause”, which means that we visited facilities where infestations were suspected or initially detected by the educational staff. This methodological approach was necessitated by the widespread perception of direct examination of children’s heads expressed by the parents and might have influenced (overestimated) the obtained result of the overall pediculosis capitis prevalence. Another factor that might have influenced the prevalence rate was the accepted criteria for the positive identification of an infestation. As a positive sign of infestation, we counted both: the presence of live lice and/or the presence of eggshells/nits found alone. Therefore it is possible that the infestation rate was overestimated slightly, because if the parents had used an effective treatment, they may not have removed the eggs, and therefore, a “false” infestation diagnosis could have been made by the nurse. However, as the same methodology has been used by our nurses before and during the COVID-19 outbreak, the data obtained are comparable and valuable from an epidemiological perspective.

It is also known that the studied population’s socio-economic status and other underlying conditions may influence the head lice infestation rate [24,27,35]. In our study, we did not evaluate this factor as this would have required obtaining additional consent and information directly from the parents of the children examined. Our study did not include direct interaction with the children’s caregivers for logistical and legal reasons, especially during the COVID-19 pandemic. 

To the best of our knowledge, this study represents the first attempt to assess the influence of COVID-19 pandemic restrictions on the prevalence of head lice infestation among children using direct examination as the primary method of investigation, particularly on such a large population of children. The findings of this study revealed a statistically significant association between the prevalence of head lice infestation among the examined children and the period of the COVID-19 pandemic in Poland. The results of this study contribute to the growing body of knowledge on the effects of pandemic restrictions on various health issues, including pediculosis capitis. They highlight the importance of implementing effective measures to control head lice infestations and emphasize the need for ongoing monitoring and awareness efforts to address this persistent epidemiological concern.

## 5. Conclusions

The restrictions implemented during the COVID-19 pandemic, which limited interpersonal contact, had an impact on the spread of head lice infestations among the tested groups of children. However, it remains unclear whether factors such as the separation of children into smaller groups or increased parental attention during lockdown contributed to the reduction in infestation rates compared to the period prior to the COVID-19 outbreak. The data obtained from our study affirm that head lice infestations continue to be an ongoing epidemiological concern that requires monitoring. Due to the lack of a standardized data collection system for tracking the prevalence of pediculosis capitis in Poland, periodic inspections of children in schools and kindergartens, similar to the ones carried out in this study, can serve as an important source of epidemiological data. 

## Figures and Tables

**Figure 1 jcm-12-04819-f001:**
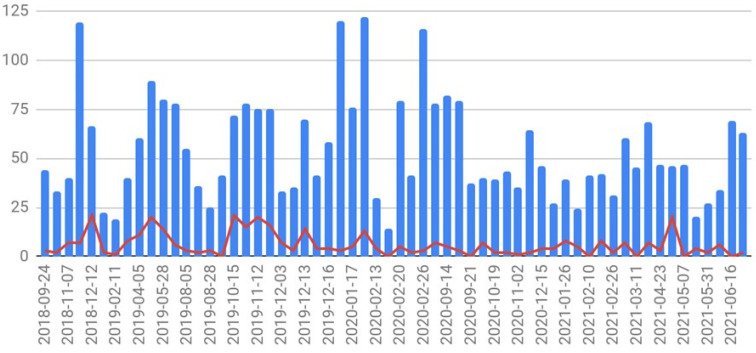
Number of children screened (blue columns) and children infested with head lice (red line) in kindergartens at each setting between 24 September 2018 and 29 June 2021.

**Figure 2 jcm-12-04819-f002:**
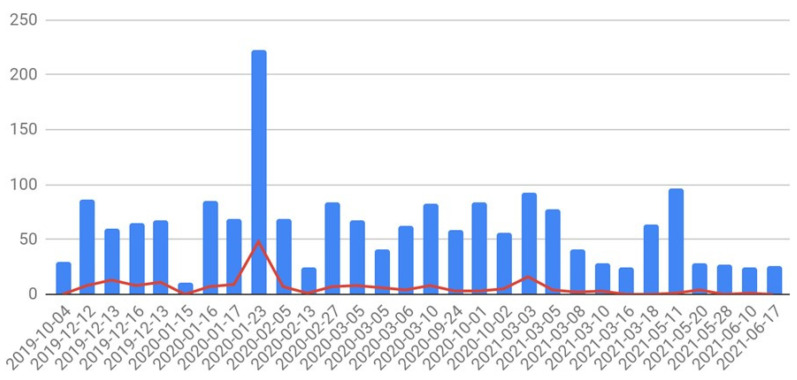
The number of children screened (blue columns) and children infested with head lice (red line) in schools at each setting between 4 October 2019 and 17 June 2021.

**Table 1 jcm-12-04819-t001:** Statistical analysis of the number of infested children and the prevalence (%) depending on the period of the COVID-19 pandemic in schools and kindergartens using the 2-tailed Fisher’s exact test with probability set at the level of *p* < 0.05.

	Kindergartens			Schools		
	Children Infested	Children not Infested	TOTAL	Children Infested	Children not Infested	TOTAL
Prior to the COVID-19 outbreak	248 (12.03%)	1812	2060	107 (10.41%)	920	1027
During the COVID-19 outbreak	116 (9.7%)	1079	1195	39 (5.37%)	687	726
*p*-value	0.0435		3255	0.0001		1753

## Data Availability

The datasets used and/or analyzed during the current study are available from the corresponding author upon reasonable request.

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
