# Peer review of "The Impact of the COVID-19 Pandemic on the Prevalence of Head Lice Infestation among Children Attending Schools and Kindergartens in Poland"

_jcm, 2023, doi:10.3390/jcm12144819_

Round 1

Reviewer 1 Report (Previous Reviewer 1)

No further comment.

Author Response

Dear Reviewer,

Thank You for your review and favorable decision.

Sincerely yours,

Marcin Padzik

Reviewer 2 Report (Previous Reviewer 3)

This is a much improved manuscript now that the changes have been made.  I especially agree with the decision to use Fisher's Exact Test for analysis because it gives a clearer basis for determining the differences between categorical values, especially if the cohort of schools post-Covid differed from the cohort examined pre-Covid because direct school to school comparisons could not have been made.

The methodology is now clearer.  My only comment is that I still find the Introduction a little heavy on information about point prevalence studies in other unrelated countries (lines 39-53) rather than concentrating specifically on Poland but that is my choice.  Otherwise the text is clear and informative.

Author Response

Dear Reviewer,

Thank you very much for your review and positive feedback. Please be informed that in other review that we received we have been asked to enrich our introduction with other aspects of pediculosis capitis such as treatment, symptoms, diagnosis etc. We tried to find a "golden mean" to facilitate all reviews, so we decided to leave as it is. Hope you will support our decision.

Sincerely yours,

Marcin Padzik

Reviewer 3 Report (New Reviewer)

The present study describes pediculosis in children of ages 3 to 14 years old in Poland during the period 24 September 2018 to 29 July 2021. The authors aimed to assess the prevalence of Pediculus humanis capitis before and after the COVID-19 pandemic among two age groups, seeking to make a comparison. In total 5008 children were examined by nurses. The topic is interesting and the authors tried to delineate it. However, the study has some limitations, explained below.

Authors should rephrase some parts of the manuscript to better outline the research question, literature search and their findings. In particular, the results are poor and should be enriched while the statistical terms should be used correctly. When possible, authors should refer to the most recent and relevant literature. I would suggest the authors to provide more information on the following:

1)      A description of the disease in humans (eg. signs, diagnosis and treatment).

2)      A reference to other species of Pediculs affecting humans since P. capitis is not the only species parasitizing humans.

3)      It would be interesting and would add value to the MS, if authors described vector-borne diseases transmitted by lice, as the topic refers to public health.

Specific comments

Line 15: There is at least one article that describes P. capitis in examined children. Please rephrase accordingly.

Line 25: Please use more fitting references.

Lines 41-42: Please provide a reference.

Lines 60-74: There is a study titled: "Head pediculosis in schoolchildren in the eastern region of the European Union". Please cite it, search for any similar studies that you may have missed and reform your text.

Lines 90-93: Please rephrase.

Lines 108-113: Please mention the keys used for the identification of the parasites and provide the relevant reference(s). Also, provide some more details on the nurses that participated in the study: Were they the same nurses examining all the children, did they receive any special guidance, did they use any specific keys for the identification, etc.

Line 119: “Results”: as stated in the general comments, this session is poorly presented and must be enriched with percentages, statistical terms etc.

Lines 142-146: Please enrich the table according to the previous comment.

Lines 166-176: Please check carefully the references and correct them.

Line 192: “In the presented study”: Please change to “present”.

Lines 197-200: Please make more apparent what the “specificity of the tested population” and “the extent of restrictions” mean and discuss.

Line 201: “review”: Please use another word. This is not a review article.

Line 215: “overestimated slightly”: Did authors take into account a possible false positive or negative identification by nurses? Please discuss the potential misidentification.

Lines 235: “especially during times of social restrictions”: According to your results, social restriction reduced the prevalence of pediculosis. Effective measures are needed to control infestations when people come in contact. Please rephrase the sentence accordingly

Lines 90-93: Please rephrase.

Line 192: “In the presented study”: Please change to “present”.

Author Response

Dear Reviewer,

Thank you very much for your very detailed review. Please find our comments in the attached file.

Sincerely Yours,

Marcin Padzik

This manuscript is a resubmission of an earlier submission. The following is a list of the peer review reports and author responses from that submission.

Round 1

Reviewer 1 Report

Major problems:

1. The sampling method most probaly biased the results. First, visits to schools and kindergartens were performed “for cause”. This means that reseachers did nor visit schools and kindergartens where infestations were not detected or suspected. Therefore, prevalences were overestimated. 

This bias was probably similar before and after the outbreak, thus the prevalence values can still be compared. But hese values are not comparable to other results from other countries, collected by other methods. This problem should be mentioned at least in the Discussion. 

2. The statistical analysis is wrong and so is the figure.

You used Mann-Whitney U-test to compare two samples (before and after), but you have not told what are the elements of these samples? Were the elements of the two samples the prevalence values obtained on the dates illustrated on the figures? One prevalence value for one date? Why? What was the sample sizes in this statistical test? Why making a statistical comparisons on the daily descriptive indices (prevalences) rather than using the raw data?

Or have I totally misunderstood your statistics?  

 Consider this:

Kindergarten, before covid: 248 infested, 1812 unifested

Kindergarten, after covid outbreak: 116 infested, 1079 unifested

Fisher's exact test 2-tailed P=0.0435

School, before covid: 107 infested, 920 unifested

School, after covid outbreak: 39 infested, 687 unifested

Fisher's exact test 2-tailed P=0.0001

3. Figure

You say "The black vertical line marks the boundary..."

But these figure contain no vertical lines, rather, these are huge black columns pointing at the maximum values on the vertical scale. Delete these black columns. It is easy to draw a vertical line in excel.

But I would recommend to use a "doughnut" type of charts, without representing the dates of collections.   

For example:

"Those are wingless obligatory blood-sucking insects that are found on the human scalp during all their lives."

correct to:

"They are wingless, blood-sucking insects living a parasitic way of life in the human scalp during their whole life cycle."

"On the Polish ground the recently performed studies ..."

correct to:

"In Poland, the recently performed studies ..."

Author Response

Dear Reviewer 1 please find our responses in the attached file. All corrections have been highlighted in yellow in the revised manuscript.

Reviewer 2 Report

Dear authors, I have read with interest your manuscript and I consider it quite good, however, you should consider the following:

Abstract:

There is too much irrelevant information (lines 11-16). Lines 19-20 should go after methods. 

Introduction:

This entire section has been devoted to reviewing concepts of pediculosis and children, however there is no context of the research question. The authors should attempt to provide the basis for their study, which has to do with the context of isolation and school closures. I suggest using this reference for support. https://doi.org/10.54034/mic.e1480

Methods:

You should avoid giving numbers of results in this section. Supposedly the total number of participants is known after the study has been applied and not before.

Results:

Addressed and adequately presented. Improve the quality of the figures.

Discussion:

As in the introduction section, we need the authors to break down their results and try to answer questions, such as why there are changes in schools and not in kindergartens. 

There should be a limitations section.

Conclusion: It should address the research question. The rest can be presented in the discussion.

The work is fairly well translated

Author Response

Dear Reviewer 2 please find our responses in the attached file. All corrections have been highlighted in yellow in the revised manuscript.

Reviewer 3 Report

The majority of this manuscript is well presented, although it is my opinion that the Introduction is unnecessarily extensive giving rather more general information that is appropriate for the particular subject of the investigation.  Whether this requires editing down to make it more specific to the study subject is a matter for discussion between authors and editors of the journal.

There are some minor points of English usage that could be improved.

A similar comment could be made about the Discussion section describing the findings of other authors on this topic and giving slightly more detail than is required for a comparison between this study, which is an important advance because it directly examines the infestation status of children rather than using “remote” methods as used by other studies.

It is not clear whether the criteria for determining an infestation was based on finding of live lice, live lice and eggs/nits, or whether the presence of eggshells/nits alone was an accepted criterion for identification of an infestation.  If the latter, it is possible that the infestation rate has been over estimated, particularly in the post-Covid period because if parents had used an effective treatment during the Covid lockdown they may not have removed eggs and so a “false” infestation diagnosis could have been made.  This needs clarification.

There are two points in the Conclusion that are contentious. 

The first is the assumption that “It may be concluded that physical distancing may reduce transmission of pediculosis capitis.”  There is no evidence of this from the recorded observations.  It is just as likely that during the lockdown periods, when schools did not operate, that parents spent more time examining their children and treating them for pediculosis with the result that when the nurses subsequently examined them the infestation had been eliminated.

The second is that from my reading of the manuscript the following statement “Presented results suggests also that the main route of the spread of head lice infestation among children in Poland is close contact between peers, rather than between family members in households.” is not supported by any of the data presented in the Results section or in the Discussion of the observations.

The final conclusion, that regular inspections in schools for control of pediculosis are required is not justified by any evidence from around the world.  In USA school inspections take place on a regular basis yet, by the estimates cited in the manuscript, head lice are still prevalent and widely distributed there.  If the inspection regimen was at all effective there should be few if any infestations so the numerical data cited on numbers of cases suggest that the school inspection regimen has been a total failure in USA.  Perhaps a better statement could be devised.

There are a few sentence constructions and slightly unusual words that show this article has been written by an author who is not an English native speaker.  Changes are not necessary but if the authors were to have the text gone over by a native speaker it would read slightly better in places.

Author Response

Dear Reviewer 3, 

Please find our detailed updates in the attached file.

Sincerely yours,

Marcin Padzik

Round 2

Reviewer 1 Report

this is much better.

Author Response

Dear Reviewer 1, thank you for your kind approval.

Reviewer 2 Report

The suggestions given have not been considered. The introduction still does not give a context to the study. In the methodology they now give another data, from where 5008 children appeared, because they were not included in the study. The limitations of your study go beyond the type of sampling (which in itself is quite debatable about the validity of the data), what about children with underlying conditions, is the socio-economic factor similar in both periods, etc.?

The conclusions are not adequately addressed.

None

Author Response

Dear Reviewer 2, 

Please find our detailed updates in the attached file.

Sincerely yours,

Marcin Padzik
